# Revisiting Sparse Mixture of Experts for Resource-adaptive Federated Fine-tuning Foundation Models

**Van-Tuan Tran**[1*]**, Khiem Le**[2*]**, Quoc-Viet Pham**[1]

[1] Trinity College Dublin, Ireland, [2] University of Notre Dame, IN, USA

[*] Equal contribution

{tranva, viet.pham}@tcd.ie

kl3@nd.edu

## Abstract

Existing federated fine-tuning methods for large-scale foundation models (FMs) assign heterogeneous low-rank adaptation (LoRA) ranks for clients based on their computation capabilities to address system heterogeneity. However, these approaches require merging LoRA matrices into the original model to obtain the full model, causing the computational overhead for resource-constrained clients at inference time. Moreover, their performance is not as effective as that of the homogeneous LoRA, in which the lowest rank is applied to all clients. To overcome these limitations, we propose a resource-adaptive federated fine-tuning method by revisiting the conditional computation property of Sparsely-activated Mixture-of-Experts (SMoE). The key principle here is to extend the data-conditional computation property of SMoE to a new dimension - **resource-conditional computation**, where clients can activate a suitable number of experts depending on their available resources. Furthermore, to address the imbalanced expert utilization caused by heterogeneous expert activation patterns, we propose a new **A**ctivation-**a**ware **a**ggregation algorithm for SMoE ($A^3$SMoE). This algorithm enhances the aggregation process by incorporating client-specific expert activation patterns. Through experiments across independent and identically distributed (IID) and non-IID scenarios, we demonstrate that our proposed method achieves superior performance compared to both homogeneous- and heterogeneous-LoRA approaches under different computation budgets. We also show that LoRA-based methods can be improved when integrated with $A^3$SMoE.

## 1 Introduction

Fine-tuning foundation models (FMs) directly for specialized domain-specific tasks Thirunavukarasu et al. (2023); Dong et al. (2023) present two significant challenges: 1) the substantial computational requirements for fine-tuning and 2) the need for extensive task-relevant data. Numerous Parameter-efficient Fine-tuning (PEFT) methods have been proposed, notably the low-rank adaptation (LoRA) approach Hu et al. (2021), to significantly reduce the number of trainable parameters and mitigate the gap in computational power Han et al. (2024). Despite effectiveness, existing PEFT methods often assume that task-relevant data are stored on a single centralized machine with enough data and resources to fine-tune FMs for adaptation to the downstream task Cho et al. (2024). In practice, data for downstream tasks is often scarce and distributed across multiple devices and is often privacy-sensitive, such as law-related documents or medical profiles, making centralized storage impractical or prohibited.

Federated fine-tuning has emerged as a promising solution, enabling distributed clients to collaboratively fine-tune FMs while preserving data privacy Zhang et al. (2024). Recent advances have focused on tackling system heterogeneity in federated fine-tuning by assigning heterogeneous LoRA ranks for clients based on their computation capabilities Wang et al. (2024); Bai et al. (2024); Cho et al. (2024). Despite the achieved progress, at inference time, LoRA-based methods require merging the low-rank matrices into the original model to obtain the full model, causing computational

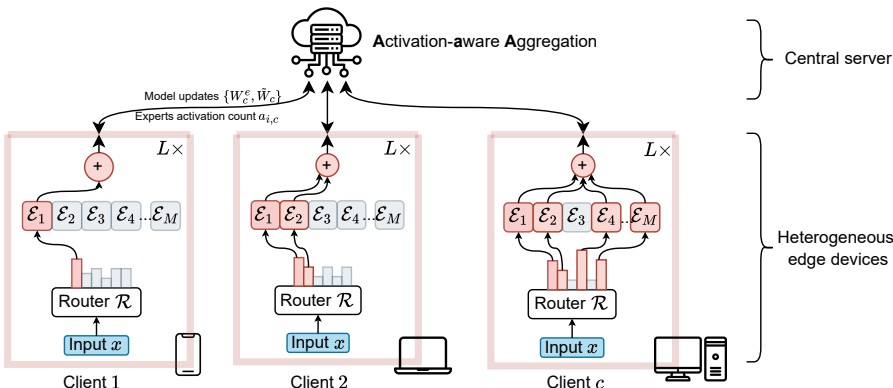

Figure 1: Overview of $A^3$SMoE, for resource-adaptive federated fine-tuning FMs, where clients activate the number of experts depending on their computational capabilities. These client-specific expert activation patterns are utilized on the server side for aggregation following Eq. (7).

overhead for resource-constrained clients. Furthermore, through our experiment, the performance of these approaches is not as effective as the lower bound, which is the homogeneous LoRA with the clients' lowest rank applied. To address these issues, we present an orthogonal approach for federated fine-tuning FMs, which addresses system heterogeneity during both training and inference phases and can be orthogonal to traditional LoRA-based methods. Specifically, we revisit the conditional computation property of Sparsely-activated Mixture of Experts (SMoE) by extending the *data-conditional computation* property to a new dimension - *resource-conditional computation*. While traditional SMoE uses a gating network to route inputs to specialized experts based on input characteristics Shazeer et al. (2017), our method introduces an additional condition of computation, where resource-constrained clients can process inputs by activating fewer experts and more capable clients can utilize a larger expert subset. Specifically, as illustrated in Figure 1, the number of top-K active experts for each client is selected based on their available resources, while the gating network determines the specific expert activation patterns according to input features. However, heterogeneous expert activation patterns across clients can lead to imbalanced expert utilization, where certain experts receive frequent updates while others remain underutilized, leading to poor expert scalability Chen et al. (2023) (illustrated in Figure 2). To mitigate this, we introduce a new activation-aware aggregation algorithm for SMoE, named $A^3$SMoE, that incorporates client-specific expert utilization patterns into the aggregation process, ensuring more balanced model updates.

Our key contributions are summarized as follows:

- We introduce a new conditional computation property during training and inference phases for federated fine-tuning FMs subject to both input data and resource availability based on SMoE. This property effectively handles system heterogeneity by enabling clients to activate different numbers of experts based on their resource capability.

- We propose $A^3$SMoE method that enables *resource-adaptive* expert activation patterns across clients, complemented by an activation-aware aggregation algorithm that considers client-specific expert utilization patterns.

- We evaluate $A^3$SMoE method on the instruction-tuning task, to show that it outperforms existing heterogeneous LoRA approaches. Furthermore, $A^3$SMoE demonstrates its orthogonal feature with significant performance improvement when implemented on top of existing heterogeneous LoRA methods.

## 2 PRELIMINARIES

### 2.1 SPARSELY-ACTIVATED MIXTURE-OF-EXPERTS

Given a Transformer Vaswani et al. (2017) model consisting of $L$ fully-connected (FC) layers, SMoE-based models enable data-conditional computation by replacing these FC layers with SMoE layers, where only a subset of experts is activated at a time Fedus et al. (2022); Chen et al. (2023).

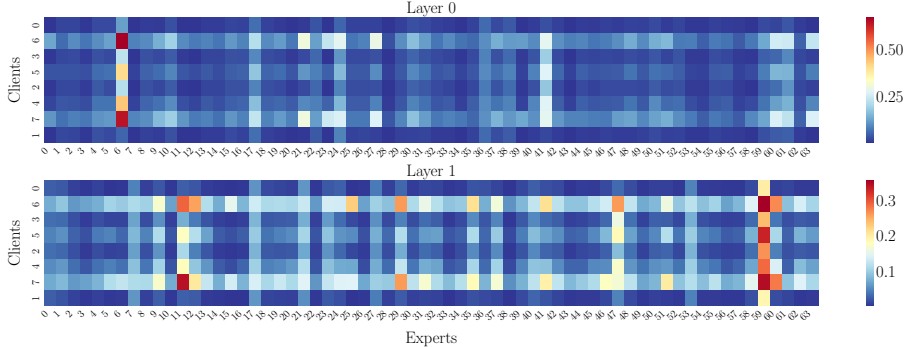

Figure 2: Illustration of heterogeneous experts utilization across clients.

**Definition 2.1.** (SMoE Layer Components). An SMoE layer consists of: 1) A set of $M$ experts $\{\mathcal{E}(\cdot; W_i^e)\}_{i=1}^M$, where each expert $\mathcal{E}(\cdot; W_i^e) : \mathbb{R}^d \to \mathbb{R}^d$ is parameterized by weights $W_i^e$; 2) A router $\mathcal{R}(\cdot; W^r) : \mathbb{R}^d \to \mathbb{R}^M$ parameterized by $W^r$; and 3) A sparsity hyperparameter $K \ll M$ controlling expert activation.

Given an input embedding $x \in \mathbb{R}^d$, the router first computes the affinity scores $s \in \mathbb{R}^M$ between $x$ and each expert through a linear transformation: $s = \mathcal{R}(x; W^r) = W^r x$, i.e., assessing each expert's relevance to the input. Then, the top-$K$ relevant experts are selected through a sparse gating function TopK $: \mathbb{R}^M \times \mathbb{Z}^+ \to \mathbb{R}^K$. Selected experts compute their outputs $y \in \mathbb{R}^d$ by the routing policy as follows:

$$y = \sum_{i=1}^M \text{softmax}(\text{TopK}(s, K))_i \mathcal{E}(x; W_i^e), \tag{1}$$

where $\text{TopK}(v, K)_i$ is defined as $v_i$ if $v_i \in \mathcal{S}_K(v)$ and $-\infty$ otherwise. Here, $\mathcal{S}_K(v) = \{i \in v : |\{j \in v : j > i\}| < K\}$ is the set of $K$ largest elements of $v$.

## 2.2 FEDERATED FINE-TUNING WITH LORA

**Definition 2.2.** (**Low-Rank Adaptation** Hu et al. (2021)) For a pre-trained FM with weight matrix $\mathbf{W}_0 \in \mathbb{R}^{d \times l}$, the weight is updated via two trainable low-rank decomposition matrices $\mathbf{B} \in \mathbb{R}^{d \times r}$ and $\mathbf{A} \in \mathbb{R}^{r \times l}$: $\mathbf{W} = \mathbf{W}_0 + \frac{\alpha}{r}\mathbf{BA}$, where $r \ll \min\{d, l\}$ is the rank of the decomposition and $\alpha$ is the scaling factor. The original pre-trained weight $\mathbf{W}_0$ remains frozen while only $B$ and $A$ matrices are updated during training.

**Definition 2.3.** (**Federated Fine-tuning with Homogeneous LoRA**) Let $\{(x, y) \in \mathcal{X} \times \mathcal{Y}\}$ denote the space of input-output pairs, where $\mathcal{X}$ is the input space and $\mathcal{Y}$ is the output space. Given $C$ clients, where each client $c \in [C]$ has local data $\mathcal{D}_c = (\mathbf{x}_i, \mathbf{y}_i)_{i=1}^N \subseteq \mathcal{X} \times \mathcal{Y}$ and its local optimization function $F_c(\mathbf{W}) = \frac{1}{|\mathcal{D}_c|} \sum_{(\mathbf{x}, \mathbf{y}) \in \mathcal{D}_c} \ell(\mathbf{W}, (\mathbf{x}, \mathbf{y}))$. Federated fine-tuning with homogeneous LoRA solves the optimization problem:

$$\min_{\mathbf{B}, \mathbf{A}} F(\mathbf{W}) = \frac{1}{C} \sum_{k=1}^C F_c(\mathbf{W}), \quad \text{s.t.} \quad \mathbf{W} = \mathbf{W}_0 + \frac{\alpha}{r}\mathbf{BA}, \tag{2}$$

where $\ell(\mathbf{W}, (\mathbf{x}, \mathbf{y}))$ is the loss for model $\mathbf{W}$ at data sample $\{x, y\}$.

Federated fine-tuning using LoRA aims to collaboratively learn the global low-rank matrices $\mathbf{B}$ and $\mathbf{A}$ across all clients while keeping pre-trained weights $\mathbf{W}_0$ fixed. Clients update their local versions of $\mathbf{B}$ and $\mathbf{A}$ matrices using their local data, and these updates are aggregated at a central server.

**Definition 2.4.** (**Federated Fine-tuning with Heterogeneous LoRA**) Let $\beta^{(c)} \in [0, 1]$ denote the computation budget for each client $c$, which represents its available computational and memory resources. The heterogeneous LoRA optimization problem is defined as follows:

$$\min_{\mathbf{B}_c, \mathbf{A}_c} F(\mathbf{W}) = \frac{1}{C} \sum_{c=1}^C F_c(\mathbf{W}_c), \quad \text{s.t.} \quad \mathbf{W}_c = \mathbf{W}_0 + \frac{\alpha}{r_c}\mathbf{B}_c\mathbf{A}_c, \tag{3}$$

where $\mathbf{B}_c \in \mathbb{R}^{d \times r_c}$, $\mathbf{A}_c \in \mathbb{R}^{r_c \times l}$, $r_c = \lfloor r_{\max}\beta^{(c)} \rfloor$ ($r_{\min} \leq r_c \leq r_{\max}$), and $r_c$ is client $c$'s LoRA rank determined by its capability factor $\beta^{(c)}$ and is bounded between the minimum rank $r_{\min}$ and the maximum rank $r_{\max}$.

# 3 METHODOLOGY

## 3.1 RESOURCE-ADAPTIVE FEDERATED FINE-TUNING SMoE

Consider a federated learning (FL) system with $C$ clients collaboratively fine-tuning an SMoE-based FM, where FC layers are replaced with SMoE layers. The global model parameters consist of expert parameters $\{\mathbf{W}_i^e\}_{i=1}^M$ and non-expert parameters $\tilde{\mathbf{W}}$, which include both router parameters and non-MoE layer parameters.

**Resource-Adaptive SMoE**. To handle system heterogeneity in federated fine-tuning, we extend the conditional computation property of SMoE into a new dimension - **resource-conditional computation**. In particular, we introduce heterogeneous sparsity by varying the sparsity level $K$ based on clients' resources. Let $\beta^{(c)} \in [0, 1]$ denote the computation budget of each client $c$, for the input embedding $x$, client $c$ computes its output as follows:

$$y_c = \sum_{i=1}^M \text{softmax}(\text{TopK}_c(s, K_c))_i \mathcal{E}(x; \mathbf{W}_i^e),\qquad(4)$$

where $\text{TopK}_c$ selects $K_c$ experts based on the client's computational budget and $K_c$ is the client-specific number of experts which is determined as $K_c = \lfloor K_{\max}\beta^{(c)} \rfloor$ and bounded by $K_{\min} \leq K_c \leq K_{\max}$. Eq. (4) indicates that the number of activated experts $K_c$ is determined by each client's computation capability via $\beta^{(c)}$.

**Heterogeneous Federated Optimization**. Each client $c$ receives the current global model parameters $W^{e,(t)}$ and $\tilde{W}^{(t)}$ and performs local fine-tuning through $\tau$ steps of stochastic gradient descent:

$$\{\mathbf{W}_c^{e,(t+1)}, \tilde{\mathbf{W}}_c^{(t+1)}\} = \text{LocalFineTune}(\mathbf{W}^{e,(t)}, \tilde{\mathbf{W}}^{(t)}, \mathcal{D}_c, \tau).\qquad(5)$$

In the heterogeneous setting, the federated optimization problem is defined as follows:

$$\min_{\mathbf{W}^e, \tilde{\mathbf{W}}} F(\mathbf{W}^e, \tilde{\mathbf{W}}) = \frac{1}{C} \sum_{c=1}^C F_c(\mathbf{W}_c^e, \tilde{\mathbf{W}}_c),\qquad(6)$$

where $F_c$ is client c's local objective computed using the resource-adaptive SMoE defined in Eq. (4).

## 3.2 ACTIVATION-AWARE AGGREGATION

While resource-adaptive federated fine-tuning with SMoE effectively addresses system heterogeneity by allowing clients to select different numbers of experts ($K_c$) based on their computational capabilities, this approach introduces a critical challenge: heterogeneous expert utilization (depicted in Figure 2). When clients activate different sets of experts during local training, some experts may be frequently updated while others remain underutilized. This imbalance can lead to sub-optimal model performance and poor expert scalability Chen et al. (2023).

To tackle this challenge, we propose an activation-aware aggregation algorithm, namely $\text{A}^3\text{SMoE}$, which incorporates the normalized utilization frequency of each expert into the aggregation mechanism. Particularly, at each communication round, the client-specific updates $\{\mathbf{W}_c^e, \tilde{\mathbf{W}}_c\}$ and client-specific experts activation counts $a_{i,c}$ are sent to the server for activation-aware aggregation (depicted in Figure 1). On the server side, different aggregation strategies are employed for expert and non-expert parameters. For expert parameters $\mathbf{W}^e$, the aggregation incorporates both the client's data size $|\mathcal{D}_c|$ and the activation count $a_{i,c}$ for expert $\mathcal{E}_i, i \in [M]$. Note that, the activation count $a_{i,c}$ is tracked layer-wise and the layer notation is removed for simplicity, without losing generality. The normalized clients' data size $p_{d_c}$ and expert activation frequency $p_{i,c}$ would be then calculated as $p_{d_c} = |\mathcal{D}_c|/\sum_{u=1}^C |\mathcal{D}_u|$ and $p_{i,c} = a_{i,c}/\sum_{j=1}^M a_{j,c}$, respectively. The activation-aware aggregation

for expert parameters is defined as follows:

$$\mathbf{W}_i^{e,(t+1)} = \sum_{c\in[C],i\in[M]} p_{d_c}\left(\frac{p_{i,c}}{K_c}\right)^{\gamma} \mathbf{W}_c^{e,(t)},\tag{7}$$

where $\gamma$ is a balancing hyperparameter. For non-expert parameters $\tilde{\mathbf{W}}$, the data-size-weighted averaging aggregation McMahan et al. (2017) is employed as follows:

$$\tilde{\mathbf{W}}^{(t+1)} = \sum_{c\in[C]} p_{d_c}\tilde{\mathbf{W}}_c^{(t)}.\tag{8}$$

By considering client-specific expert activation counts into aggregation, $\text{A}^3\text{SMoE}$ can help mitigate the imbalance of expert utilization.

Table 1: Comparison with LoRA-based methods using OLMoE-1.3B-7B model.

| | Computation budget | IID ($\alpha = 5.0$) | | | | Non-IID ($\alpha = 0.5$) | | | |
|---|---|---|---|---|---|---|---|---|---|
| | | Homo-LoRA | HetLoRA | FlexLoRA | $\text{A}^3\text{SMoE}$ | Homo-LoRA | HetLoRA | FlexLoRA | $\text{A}^3\text{SMoE}$ |
| 8 clients | $\beta_1$ (0.6B) | 0.0777 | 0.0659 | 0.0690 | **0.2255** | 0.0818 | 0.0648 | 0.0657 | **0.2157** |
| | $\beta_2$ (0.7B) | 0.2400 | 0.1845 | 0.2091 | **0.2769** | 0.2350 | 0.1863 | 0.2019 | **0.2855** |
| | $\beta_3$ (0.9B) | 0.3201 | 0.3041 | 0.3144 | **0.3301** | 0.3146 | 0.2987 | 0.3068 | **0.3294** |
| | $\beta_4$ (1.3B) | 0.3347 | 0.3275 | 0.3361 | **0.3414** | 0.3296 | 0.3252 | 0.3327 | **0.3397** |
| 40 clients | $\beta_1$ (0.6B) | 0.0673 | 0.0619 | 0.0628 | **0.1885** | 0.0648 | 0.0627 | 0.0645 | **0.1927** |
| | $\beta_2$ (0.7B) | 0.2203 | 0.1372 | 0.1432 | **0.2608** | 0.2290 | 0.1289 | 0.1476 | **0.2588** |
| | $\beta_3$ (0.9B) | 0.2948 | 0.2731 | 0.2955 | **0.3059** | 0.2951 | 0.2742 | 0.2843 | **0.3012** |
| | $\beta_4$ (1.3B) | 0.3111 | 0.3101 | **0.3231** | 0.3156 | 0.3138 | 0.3055 | 0.3141 | **0.3152** |

Table 2: Comparison with LoRA-based methods using dense model (OLMo-1.3B).

| | IID ($\alpha = 5.0$) | | | | Non-IID ($\alpha = 0.5$) | | | |
|---|---|---|---|---|---|---|---|---|
| | Homo-LoRA | Het-LoRA | FlexLoRA | $\text{A}^3\text{SMoE}$ | Homo-LoRA | Het-LoRA | FlexLoRA | $\text{A}^3\text{SMoE}$ |
| 8 clients | 0.3062 | 0.3006 | 0.3056 | **0.3414** | 0.3097 | 0.2918 | 0.3012 | **0.3397** |
| 40 clients | 0.2959 | 0.2828 | 0.2891 | **0.3156** | 0.2967 | 0.2787 | 0.2866 | **0.3152** |

## 4 EXPERIMENTS

### 4.1 EXPERIMENTAL SETUP

We conduct experiments on the Instruction-tuning task for FMs on the Dolly-15k dataset Conover et al. (2023). The data is split to $8$ and $40$ clients follow the Dirichlet distribution with concentration parameter $\alpha \in \{5.0, 0.5\}$ corresponding to the heterogeneity levels of clients' data Li et al. (2022). The ROUGE-L metric Lin (2004) is adopted for evaluation. Regarding system heterogeneity, we established $4$ levels of computational budgets $[\beta_1, \beta_2, \beta_3, \beta_4]$, following the experimental setup of previous works Liu et al. (2024); Diao et al. (2021). For the SMoE model (OLMoE-1.3B-7B), $1, 2, 4, 8$ out of $64$ total experts are activated accordingly, corresponding to 0.6B, 0.7B, 0.9B, 1.3B activated parameters.

### 4.2 EXPERIMENTAL RESULTS

#### 4.2.1 $\text{A}^3\text{SMoE}$ OUTPERFORMS HOMOGENEOUS AND HETEROGENEOUS-LoRA METHODS

We evaluate the performance of $\text{A}^3\text{SMoE}$ with homogeneous and heterogeneous LoRA methods using the OLMoE-1.3B-7B Muennighoff et al. (2024) model. As shown in Table 1, $\text{A}^3\text{SMoE}$ outperforms Homo-LoRA, HetLoRA Cho et al. (2024), and FlexLoRA Bai et al. (2024) across diverse FL scenarios. The improvement is particularly noticeable in lower computation budgets ($\beta_1$ and $\beta_2$), highlighting $\text{A}^3\text{SMoE}$'s efficiency in handling computational limitations of edge devices. From

Table 1, we also observed that A³SMoE maintains high ROUGE-L scores over other LoRA-based methods when the system scales up to 40 clients, e.g., 0.1885 (IID) and 0.1927 (non-IID) at the lowest computation budget of 0.6B.

We compare A³SMoE with LoRA-based approaches using the dense model (OLMo-1.3B Groeneveld et al. (2024)) with a similar number of parameters as that of the SMoE model. The results in Table 2 show the superior performance of A³SMoE compared to other LoRA-based methods. Noticeably, Tables 1 and 2 reveal that the homogeneous LoRA method achieves comparable performance with heterogeneous-LoRA methods, including HetLoRA and FlexLoRA. This phenomenon is aligned with previous research on system heterogeneity in FL Liu et al. (2024).

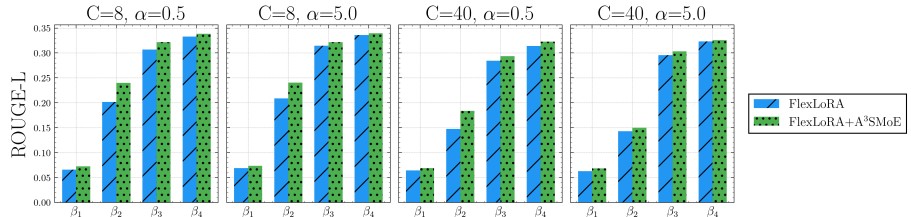

Figure 3: Performance improvement when integrating A³SMoE.

### 4.2.2 A³SMoE IMPROVES HETEROGENEOUS LORA METHODS

Beyond outperforming existing LoRA-based methods, A³SMoE also demonstrates its versatility. To be more specific, it can be effectively integrated with existing LoRA-based methods to enhance their performance further by concurrently using both heterogeneous LoRA ranks and resource-conditional SMoE for local fine-tuning. As illustrated in Figure 3, when implemented on top of FlexLoRA, A³SMoE significantly improves the performance across both IID and non-IID scenarios with different computation budgets.

### 4.2.3 ABLATION STUDY

**Effect of balancing hyperparameter** $\gamma$: We also provide an ablation study in Figure 4 to evaluate the effect of $\gamma$ values on the performance of A³SMoE across different computation budgets. While the performance improvements are consistent across $\beta$ configurations when increasing $\gamma$ from 0.0 to 4.0, the performance trends stabilize or slightly decline beyond $\gamma = 8.0$.

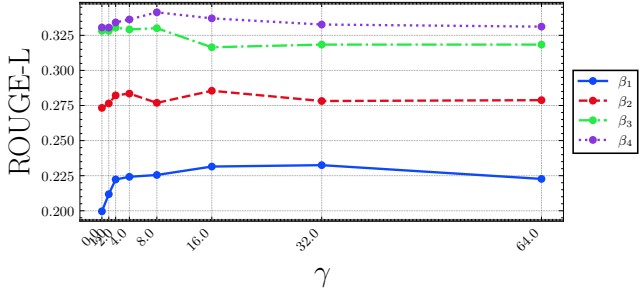

Figure 4: Performance under varying $\gamma$ values.

## 5 CONCLUSIONS

This study extends the data-conditional computation property of SMoE to a new dimension, called resource-conditional computation. Our method enables clients to activate the number of experts based on their computational capabilities. To handle the imbalanced expert utilization caused by heterogeneous expert activation patterns across clients, we propose A³SMoE, which incorporates client-specific expert activation patterns for the aggregation process on the server side. Compared with homogeneous and heterogeneous LoRA methods, A³SMoE achieves superior performance across both IID and non-IID scenarios. Significantly, A³SMoE demonstrates significant performance improvement over other LoRA-based methods in low computation budgets, showing its effectiveness in handling system heterogeneity.

## ACKNOWLEDGMENTS

This publication has emanated from research conducted with the financial support of Taighde Éireann – Research Ireland under Grant number 18/CRT/6222. The work of Quoc-Viet Pham (corresponding author) is supported in part by European Union under the ENSURE-6G project (Grant Agreement No. 101182933).

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

APPENDIX

## A    IMPLEMENTATION DETAILS

Our implementation of A$^3$SMoE uses a consistent LoRA rank of 20 across all clients, while the heterogeneous LoRA approaches (HetLoRA and FlexLoRA) employ varying ranks of $[12, 16, 24, 40]$ corresponding to the computational budgets $[\beta_1, \beta_2, \beta_3, \beta_4]$. For all LoRA-based methods, we set the scaling parameter $\alpha$ to 16. Client-side fine-tuning is performed using the Adam optimizer with a learning rate of $1.5e^{-4}$ and batch size of 16, and each client trains locally for 1 epoch per communication round. The federated training process consists of 2 communication rounds between clients and the server, with only the trainable parameters (LoRA matrices A and B) being transferred to reduce communications overhead.

## B    RELATED WORKS

### B.1    MODEL HETEROGENEITY IN FL

To enable clients to have different-sized local models depending on their available resources, existing approaches have employed model partitioning strategies that extract submodels from a global model through width-based scaling (channel pruning) or depth-based scaling (early exiting).

**Width-based Scaling**.   HeteroFL Diao et al. (2021) creates smaller submodels for resource-constrained clients by reducing the number of convolutional channels of each layer using a shrinkage ratio. FjORD Horvath et al. (2021) introduces Ordered Dropout to systematically drop adjacent channels from right to left in each layer, creating nested subnetworks where smaller models are proper subsets of larger ones. Instead of pruning channels like HeteroFL and FjORD, FLANC Mei et al. (2022) takes a different approach by representing networks as linear combinations of a shared neural basis, allowing all parameters to benefit from the complete client dataset. However, real-world deployments often require dynamic adaptation of model properties beyond just size. For example, an autonomous vehicle may need a more robust model when entering poor lighting conditions. Split-Mix Hong et al. (2022) addresses this by combining width-based scaling with dual batch normalization, which utilizes separate branches for clean and adversarial data to provide varying robustness levels. A key limitation of these approaches is their reliance on static subnetwork extraction from fixed portions of the global model. This introduces two main drawbacks: (i) The global server model is restricted to be the same size as the largest client model; (ii) Some layers can only be trained on powerful clients while others can be trained on all clients. To overcome these limitations, FedRolex Alam et al. (2022) introduces a rolling submodel extraction scheme that utilizes a rolling window to systematically extract different parts of the global model across training rounds, allowing all parts of the model to be evenly trained across all clients' data over time.

**Depth-based Scaling**. InclusiveFL Liu et al. (2022) assigns models of different sizes to clients with different computing capabilities while sharing knowledge among them through layer-wise parameter sharing. To enable effective knowledge transfer, the method uses a momentum distillation technique where the last encoder layer in smaller models imitates the behavior of corresponding layers in larger models, allowing all clients to contribute to training a large and powerful global model. DepthFL Kim et al. (2023) incorporates an additional bottleneck layer and an independent classifier for every submodel. The method also uses mutual self-distillation between classifiers within each local model, allowing shallow classifiers to train deeper classifiers. However, these depth-wise approaches exhibit two primary limitations: (i) Deep submodels in these approaches suffer from insufficient training data since they can only be trained on high-resource clients; (ii) Previous approaches rely on multiple separate classifiers, which leads to competing optimization criteria and conflicting gradients between submodels. To overcome these limitations, several works leverage both depthwise and widthwise scaling to accommodate clients with varying computational capabilities Lee et al. (2024); Ilhan et al. (2023); Kang et al. (2023).

While existing model partitioning approaches show promise, they struggle to handle heterogeneous model architectures at scale, particularly when dealing with large foundation models and their fine-tuned variants. Our method addresses this limitation by enabling federated fine-tuning of large-scale models in heterogeneous environments, in terms of clients' computational capabilities.

## B.2 RESOURCE-ADAPTIVE FEDERATED FINE-TUNING FMS

FedIT Zhang et al. (2024) first directly applied FedAvg McMahan et al. (2017) to LoRA modules in which clients fine-tune LoRA Hu et al. (2021) modules using their local data and then send the fine-tuned modules to the server. On the server, FedAvg is utilized to average all clients' LoRA modules to obtain a global LoRA. However, FedIT allocates fixed LoRA ranks to each client, which cannot satisfy the system heterogeneity nature in FL in which each client has different resource and computation capabilities. To address this limitation, FLoRA Wang et al. (2024) introduces a stacking-based noise-free aggregation method that stacks two decomposition LoRA matrices independently. To enable federated fine-tuning in the heterogeneous settings, HetLoRA Cho et al. (2024) distributes truncated versions of the global LoRA modules for clients based on their computational capability. During local training, clients can self-prune their ranks using a regularization term, reducing noise from unnecessarily large ranks. For aggregation, a sparsity-weighted scheme is introduced to de-emphasize updates from clients with larger ranks but less informative updates while giving more weight to clients with more informative updates regardless of their rank size. This helps balance the contributions between high- and low-rank clients based on the quality of their updates rather than just their rank sizes. FlexLoRA Bai et al. (2024) leverages SVD for weight redistribution and allows dynamic adjustment of local LoRA ranks based on client resources.

However, at inference time, these LoRA-based methods require merging the low-rank matrices into the original model to obtain the full model, causing computational overhead for resource-constrained clients. Furthermore, through our experiment, the performance of these approaches is not as effective as the lower bound, which is the homogeneous LoRA with the clients' lowest rank applied. In this work, we aim to handle these limitations of LoRA-based approaches by presenting an orthogonal federated fine-tuning approach. Specifically, we introduce conditional computation property during training and inference phases for federated fine-tuning FMs subject to both input data and resource availability based on SMoE.

