# OpenReview forum: "Revisiting Sparse Mixture of Experts for Resource-adaptive Federated Fine-tuning Foundation Models"
_ICLR.cc/2025/Workshop/MCDC — MCDC @ ICLR 2025_

### Official Review · Reviewer_g8s7 · 2025-02-25

**Rating:** 6
**Confidence:** 4
**Fit:** 5

**Summary:**

The paper proposes a federated fine-tuning method suitable for networks with system heterogeneity. The proposed method is based on the Sparsely-Activated Mixture of Experts (SMoE) model, where the sparsity level (i.e., the number of activated experts) depends on each client’s computational resources. One technical challenge in training this model is the imbalance in expert updates, as some experts may be updated more frequently than others. To address this, the authors propose an activation-aware aggregation step (activation refers to the active experts here) that accounts for client-specific updates along with client-specific expert activation counts. Instead of the standard averaging step, the proposed aggregation computes a weighted average of expert parameters where the weights depend on the activation counts. The proposed method is shown to outperform other baseline methods for this problem through numerical experiments with the Dolly-15k dataset.

**Reason For Giving A Higher Score:**

Please see strengths.

**Reason For Giving A Lower Score:**

Please see weaknesses.

**Strengths And Weaknesses:**

Strengths:
- Perfect fit for the workshop theme
- Well written
- Promising empirical performance of the proposed method

Weaknesses:
- Heuristic nature of the proposed approach
- Very limited numerical experiments (only one dataset, only one splitting, etc.)
- Missing details on the numerical experiments (important details like number of local steps, the stopping condition for each method, are the results averaged over random seeds, etc.) -- the results are not reproducible unless the authors decide to share the code

About the novelty:
Novel in the sense this model has not been applied before to address system heterogeneity
Limited novelty in the sense the method simply mimics existing techniques to deal with data heterogeneity.

**Suggestions:**

Given that the study relies on empirical analysis, a more comprehensive and carefully designed set of numerical experiments is important to reach a clear conclusion when reading the paper.

---

### Official Review · Reviewer_HrDf · 2025-02-27

**Rating:** 7
**Confidence:** 4
**Fit:** 5

**Summary:**

The authors propose resource-conditional SMoE, which supports clients activate a suitable number of experts based on their available resources. This aggregation process is enhanced by incorporating client-specific expert activation patterns.  Empirical stidies show the proposed method achieves superior performance in various scenarios.

**Reason For Giving A Higher Score:**

The method provides a general approach to solve the system heterogeneity issue in federated learning. Overall the work is interesting and is likely to be deployed in production. Authors have also demonstrated the ability of the proposed method to work well with the existing LoRA methods.

**Reason For Giving A Lower Score:**

The clarity of the exposition can be improved by adding an algorithm block. Also, the relation between LoRA and SMoE is somewhat weak.

**Strengths And Weaknesses:**

Strength:
1. The motivation of this work is clear and the studied problem on system heterogeneity is important.
2. The experiments are extensive in the sense that various situations including different budget levels, different client numbers and non-i.i.d data distributions, are considered.
3. The proposed method can be effectively integrated with existing LoRA-based methods.

Weakness:
1. The role of LoRA in the proposed SMoE paradigm is not clear. LoRA is involved only in the local finetuning step (5). It seems the proposed method can be translated to other parameter-efficient fine-tuning methods, such as ControlNet[1]. What are the unique challenges brough up by LoRA in this work?
2. The  exposition of the proposed paradigm are lacking some details: The LoRA parameters are merged into experts weights after local finetuning by (7). Then, how are the LoRA parameters re-initialized in the next round?

[1] Zhang, L., Rao, A. and Agrawala, M., 2023. Adding conditional control to text-to-image diffusion models. In Proceedings of the IEEE/CVF international conference on computer vision (pp. 3836-3847).

**Suggestions:**

1. A clear algorithm block will be extermely helpful for the readers to understand the workflow of the proposed method.
2. The authors can consider to extend the methods to other finetuning methods, if applicable.
3. An intuitive explanation on the design and choice of balancing parameters in (7) is highly encouraged.

---

### Official Review · Reviewer_oz6X · 2025-02-27

**Rating:** 7
**Confidence:** 5
**Fit:** 5

**Summary:**

This paper proposes a novel approach for federated fine-tuning of large-scale foundation models by extending the sparsely-activated mixture-of-experts (SMoE) paradigm to account for resource heterogeneity. In contrast to standard LoRA methods—which either use a fixed (homogeneous) rank or assign heterogeneous ranks that necessitate expensive merging at inference—the proposed method enables each client to activate a number of experts (determined by its computational budget) during both training and inference. To address the resulting imbalance in expert utilization across clients, the authors introduce an activation-aware aggregation algorithm (A3SMoE) that weights client updates by both their local data sizes and the frequency of expert activation. Experimental results on an instruction-tuning task (using the Dolly-15k dataset) under both IID and non-IID conditions show that A3SMoE outperforms both homogeneous and heterogeneous LoRA baselines, particularly at lower computation budgets. Moreover, the approach is shown to be complementary, improving performance when integrated with existing LoRA methods.

**Reason For Giving A Higher Score:**

The work offers a novel and practically significant solution to the problem of system heterogeneity in federated fine-tuning. The extension of SMoE to handle resource constraints, combined with a thoughtful aggregation strategy that leverages client-specific activation patterns, represents a meaningful advancement. The promising empirical results across different computational budgets further support the paper’s contributions, making it a strong candidate for inclusion in the workshop.

**Reason For Giving A Lower Score:**

Despite its strengths, the paper’s experimental validation is limited to a single dataset and task, which raises questions about its generalizability.

**Strengths And Weaknesses:**

**Strengths:**
- **Novelty:** The paper introduces a compelling extension of SMoE by incorporating resource-conditional computation, addressing a critical challenge in federated fine-tuning where clients have diverse computational capabilities.
- **Activation-aware Aggregation:** The proposed aggregation strategy that factors in client-specific expert activation counts is innovative and effectively tackles the imbalance issue in heterogeneous expert utilization.
- **Empirical Validation:** The experimental evaluation is thorough, covering multiple computation budgets and both IID and non-IID data distributions, which convincingly demonstrates the advantages of A3SMoE over baseline methods.
- **Integration Potential:** The demonstration that A3SMoE can be integrated with existing LoRA-based methods to further boost performance adds to its practical significance.

**Weaknesses:**
- **Limited Dataset Scope:** The experiments are confined to a single instruction-tuning task on the Dolly-15k dataset. Broader evaluation across multiple tasks or datasets would strengthen the generality of the findings.
- **Clarity of Method Description:** Some parts of the method, particularly the details of the activation-aware aggregation mechanism, could benefit from additional clarification and more intuitive explanation.
- **Discussion of Limitations:** The paper would be improved by a more explicit discussion of potential limitations—for instance, scenarios where extreme heterogeneity might lead to issues that the current aggregation scheme does not fully address.

**Suggestions:**

- **Broaden Experimental Evaluation:** Consider testing A3SMoE on additional datasets or tasks to further validate its robustness.
- **Robustness of the Aggregation Strategy:** Elaborate on the activation-aware aggregation algorithm (Equation (7)) by discussing its behavior in edge cases—such as when certain experts are rarely activated or when client updates vary widely in quality.
- **Detailed Analysis of Expert Activation Dynamics:**  Provide a more comprehensive discussion on the trade-offs between activating different numbers of experts. For instance, include analyses or visualizations that illustrate how varying \(K_c\) impacts both computational cost and model performance. This can help readers understand the practical implications of resource-conditional computation.

---

### Decision · Program_Chairs · 2025-03-06

**Decision:**

Accept

**Comment:**

The proposed method account for resource discrepency in a FL setting where different client can have different resources. The resource-conditional SMoE allows to activate a suitable number of experts based on their available resources. Most of the reviewers liked the paper, found it relevant to the workshop, and recommended acceptace. We suggest the authors to incorporate the comments of the reviewers to further strengthen the paper. Overall, we're recommend to accept this work to the workshop.